# miR3633a-*GA3ox2* Module Conducts Grape Seed-Embryo Abortion in Response to Gibberellin

**DOI:** 10.3390/ijms23158767

**Published:** 2022-08-07

**Authors:** Yunhe Bai, Xiaowen Zhang, Xuxian Xuan, Ehsan Sadeghnezhad, Fei Liu, Tianyu Dong, Dan Pei, Jinggui Fang, Chen Wang

**Affiliations:** 1College of Horticulture, Nanjing Agricultural University, Nanjing 210095, China; 2Department of Plant Biology, Faculty of Biological Science, Tarbiat Modares University, Tehran P.O. Box 14115-111, Iran

**Keywords:** grapevine, miR3633a, gibberellin, *GA3ox2*, embryo abortion

## Abstract

Seedlessness is one of the important quality and economic traits favored by grapevine consumers, which are mainly affected by phytohormones, especially gibberellin (GA). GA is widely utilized in seedless berry production and could effectively induce grape seed embryo abortion. However, the molecular mechanism underlying this process, like the role of RNA silencing in the biosynthesis pathway of GA remains elusive. Here, *Gibberellin 3-β dioxygenase2* (*GA3ox2*) as the last key enzyme in GA biosynthesis was predicated as a potential target gene for miR3633a, and two of them were identified as a GA response in grape berries. We also analyzed the promoter regions of *genes* encoding GA biosynthesis and found the hormone-responsive elements to regulate grape growth and development. The cleavage interaction between VvmiR3633a and *VvGA3ox2* was validated by RLM-RACE and the transient co-transformation technique in tobacco in vivo. Interestingly, during GA-induced grape seed embryo abortion, exogenous GA promoted the expression of VvmiR3633a, thereby mainly repressing the level of *VvGA3ox2* in seed embryos. We also observed a negative correlation between down-regulated *VvGA20ox2*/*VvGA3ox2* and up-regulated *VvGA2ox3*/*VvGA2ox1*, of which GA inactivation was greater than GA synthesis, inhibited active GA content, accompanied by the reduction of VvSOD and VvCAT expression levels and enzymatic activities. These series of changes might be the main causes of grape seed embryo abortion. In conclusion, we have preliminarily drawn a schematic mode of GA-mediated VvmiR3633a and related genes regulatory network during grape seed abortion induced by exogenous GA. Our findings provide novel insights into the GA-responsive roles of the VvmiR3633a-*VvGA3ox2* module in the modulation of grape seed-embryo abortion, which has implications for the molecular breeding of high-quality seedless grape berries.

## 1. Introduction

Gibberellin (GA) is a class of biguanide plant hormones that play key roles throughout the plant’s life cycle, such as seedless induction, seed development and germination, stem elongation, root growth, flowering and fruit development [1,2,3,4]. In Arabidopsis, the GA synthesized in the ovule is transported to the valve, which coordinates the growth of the silique [5]. Moreover, the dynamicity of GA interaction with other hormones interferes with fruit development and maturation [6], of which the effect of GA on modulation of seed development is far much outstanding, especially in grape seedless induction.

The effect of GA is mainly dependent on the content of active GA, and the biosynthesis and metabolism of active GA is a multi-step enzymatic reaction process involving multiple enzymes, and the expression of these enzyme genes precisely regulates the rate of gibberellin synthesis and metabolism [7]. Among these enzymes, GA oxidase (GA3ox) is one of the key enzymes in the last step of the GA synthesis process, which can promote active GA synthesis [8]. Furthermore, the activity of GA3ox is regulated by the negative feedback of active GA [9]. Previous studies have shown that the loss of GA3ox affects the synthesis of active GA and results in varying degrees of dwarfing [10]. Conversely, over-expression of GA3ox increases the distance between stems of transgenic plants [11]. In addition, GA3ox is differentially expressed in different tissues, especially in flowers, seeds, pods, and fruits [9,12,13,14]. So far, the GA3oxs family has been reported in several species, such as Arabidopsis, poplar, wheat, and pumpkin [12,15,16,17], and their members lead to some discrepancy or redundancy of functions in various plant species. Besides, GA3oxs were reported to their roles in the GA synthesis pathway, and which factors could regulate them in their upstream remains elusive.

Nowadays, microRNAs (miRNAs) are a new type of small non-coding RNAs with 19–24 nucleotide (nt) in size, commonly found in eukaryotic organisms [18], which plays crucial regulatory roles in various physiological and metabolic processes, such as plant growth, signal transduction and responses to biotic and abiotic stress, by mediating target messenger RNA (mRNA) cleavage or repressing their translation in most eukaryotes [19,20]. For example, miR156 can target the SPL transcription factor gene family to regulate the transition from vegetative growth to reproductive growth [21]. In addition, miR172 can pair with the mRNA of its target gene APETALA2, inhibit its expression at the translational level, and regulate flower development [22,23]. Meanwhile, miR160, miR164, miR165/miR166, and miR167 play regulatory roles in flower and berry development [24,25]. In addition, VvmiR160s and VvmiR397a mediate VvARF and VvLACs during grape seed development, respectively [25,26], which could mediate target levels to modulate seed development by responding to GA signal in grapes.

Grape is one of the most important economic fruits with high nutritional value and is widely cultivated throughout the world. It is a highly GA-sensitive plant species. Application of exogenous GA to grape inflorescences at the bloom stage promoted flower opening, induced seed abortion, accelerated fruit coloring, and affected the shape of the grape berry. Our previous study identified 130 GA-responsive miRNAs in the grape that might be possibly specific to the grape and possess more species-specific functions [27]. Among them, VvmiR061 plays an important role in grape flower development and the gibberellin signal transduction pathway [28]. GA-DELLA(SLR1)-VvmiR159c-*VvGAMYB* is a novel signal transduction pathway that modulates parthenocarpy during GA-induced parthenocarpy [29]. Our previous study found that VvmiR3633a could be in response to GA in grape berries and predicted *VvGA3ox2* to be its potential target gene; thus, they might respond to GA signal to modulate the development of grape berries [27]. However, it remains elusive on how the VvmiR3633a-*VvGA3ox2* module could mediate seed-embryo abortion during GA-induced seedless berries in grapes, and the related mechanism is also unclear yet even in model plants like *Arabidopsis thaliana*, poplar (genus *Populus*), and *Oryza sativa*. To better recognize the molecular mechanism of this module and its regulatory network during the GA-induced grape seed-embryo abortion process, we identified and characterized VvmiR3633a and its target *VvGA3ox2* validated their interaction mechanism and elucidated this module-mediated regulatory network by responding to GA during grape embryo abortion process. Our findings provide significant insight into the VvmiR3633a-*VvGA3oxs* module by specific functions in this respect of the regulatory network of GA-induced embryo abortion process in grapes, which are essential for the modulation of the development of high-quality seedless grape berries.

## 2. Results

### 2.1. GA inhibits Berry and Seed Development and Leads to Grape Seed-Embryo Abortion

To investigate the effect of GA on ‘Fujiminori’ grape berry and seed development, morphological and physiological analyses were carried out during grape seed-embryo abortion. The transverse and longitudinal diameter of ovules decreased after GA treatment (Figure 1A,C), and the single grain mass of ovules decreased gradually compared with the control group (Figure 1D). We also found that exogenous GA could inhibit the development of ‘Fujiminori’ grape ovules. Moreover, GA causes wilting, yellowing, and necrosis of the ovule and finally leads to abortion. Meanwhile, in GA-treated plants, the transverse and longitudinal diameter of berries and single grain weight increased (Figure 1B,D). The color turning period was about 7d earlier than the control (Figure 1A). These changes reflect the potential shift of grape berry and seed development at their molecular levels.

### 2.2. Characterization of miR3633a and Its Complementary with VvGA3ox2

The mature sequence of VvmiR3633a was obtained by miR-RACE and was identical to the sequence in miRbase 22.1 (GGAATGGATGGTTAGGAGAG). VvmiR3633a was located in Chr17 (Figure 2A), in which its precise and star sequences were identified as 5′-GGAAUGGAUGGUUAGGAGAG-3′ and 5′-UUCCUAUACCACCCAUUCCCUA-3′ (Figure 2B), respectively. The precursor sequence of VvmiR3633a can be folded into a stable neck ring structure (Figure 2B). The VvmiR3633a precursor gene(*VvMIR3633a*) was amplified by gene-specific primers using the total DNA of the ‘Fujiminori’ grape ovule as a template. A specific band of about 500 bp in size was obtained after gel electrophoresis. After purification and sequencing, the *VvMIR3633a* sequence was identified as 504 bp (Figure 2C).

Based on the sequence of VvmiR3633a and combining it with the mRNA database (GRAPEIGGP 12X assembly-v2 annotation), we used the online software psRNATarget to predict the target gene of VvmiR3633a. The results showed that *Gibberellin 3-β dioxygenase2* (*VvGA3ox2*, VIT09s0002g05300) was identified as the candidate target gene of VvmiR3633a, and their interaction mode was cleavage action with a mismatch rate of 3.5 (yellow: mismatch value was 0.5, blue: mismatch value was 1) (Figure 2D).

### 2.3. Identification of VvmiR3633a-Targeted VvGA3ox2 and Its Sequence Structure and Evolution across Various Plant Species

We cloned the *VvGA3ox2* CDS sequence using the ovule of ‘Fujiminori’ as a template and identified its derived amino acid sequence. The open reading frame (ORF) of *VvGA3ox2* is 1080 bp (Figure 3A) and encodes a protein with 359 amino acid residues (Figure 3B). The subcellular localization assay of VvGA3ox2 was performed, and through *35S::VvGA3ox2-GFP* and *35S::GFP* were transiently expressed in tobacco leaf epidermal cells. We found that the fluorescence in tobacco cells transformed with *35S::VvGA3ox2-GFP* was localized in the membrane (Figure 3C). The homology alignment of amino acid sequences showed that GA3ox protein had high homology among different species and contained two conserved domains, including DION and 2OGFeIIOxy among different species (Figure 3D,E). The similarity of VvGA3ox2 in the sequences of constructors in various plant species suggests that it possesses a potentially conserved function. Moreover, we constructed a phylogenetic tree of VvGA3ox2 from its homologous protein using a full-length amino acid sequence according to a proximity-based approach (Figure 3F). Results demonstrated that GA3ox is conserved among various plant species. Totally, 21 GA3ox-related proteins were divided into three groups, of which group III contained 10 species, and VvGA3ox2 had a close phylogenetic relationship with *Rosa chinensis* (XP024184947.1) and a distance phylogenetic relationship with *Gossypium hirsutum* (XP016699373.1) and *Lactuca sativa* (XP023761805.1). The exon-intron structure analysis was performed to gain more insight into the VvGA3ox2 gene sequence constructor (Figure 3G). The VvGA3ox2 sequence with a large intron is longer than all the remaining orthologous sequences and reaches 4978 bp. Except for the *Rosa chinensis* and *Lactuca sativa* species, the other species have the same sequence structure, containing two exons, one intron, and upstream and downstream sequences of the gene.

We further carried out a conserved motif analysis of GA3ox protein sequences from 21 different species by MEME Suite (Figure 3H). The results exhibited that the motif number of these homologous sequences varied from 7 to 12 across different species, of which motif2, motif3, motif4, motif5, motif6, and motif9 were completely conserved in all 21 species, followed by motif1, motif7, motif8, and motif10 with conservation at 20 species, while motif11 and motif 12 had the less conservative. Protein structure domain analysis also showed high conservative domains. All these results suggest that VvGA3ox2 might possess a high conservative with the homologous proteins across various plant species.

### 2.4. Cis-Acting Element Analysis and Promoter Activity of VvMIR3633a and VvGA3ox2 in Response to GA

To recognize the possible functions of *VvMIR3633a* and *VvGA3ox2* in grapes, we analyzed *cis*-acting elements in their promoters. Among all *cis*-elements, except for the basic *cis*-elements, such as TATA-box and CAAT-box, the remaining *cis*-elements are classified into six categories, including photo-responsive elements, tissue-specific components, circadian rhythm elements, stress response elements, hormone response elements, and transcription factor binding site (Figure 4A). We found that a large number of light-responsive elements are enriched in *VvMIR3633a* and *VvGA3ox2* promoters. Moreover, a large number of components that respond to external environmental stress, such as ARE, LTP, and GC-motif, have been found. In contrast, *cis*-elements in response to circadian, tissue-specific, and transcription factor binding sites were less determined. Interestingly, hormone-responsive elements and tissue-specific differed in promoter regions of *VvMIR3633a* and *VvGA3ox2*. *VvMIR3633a* contained five tissue-specific elements and one hormone-related element, while *VvGA3ox2* contained one tissue-specific element and nine hormone-related elements. VvGA3ox2 is one of the key enzymes in the GA synthesis pathway; therefore, we analyzed the hormone-responsive elements in promoter’s sequences of *VvMIR3633a* and *VvGA3ox2* in detail (Figure 4A). The promoter of *VvMIR3633a* had only one ethylene-responsive element, while the promoter of *VvGA3ox2* had nine hormone-responsive elements, including 4 MeJA-, 3 Ethylene-, 1 SA-, and 1 GA-responsive element. Interestingly, both *VvMIR3633a* and *VvGA3ox2* contained ethylene-responsive elements indicating that both might directly respond to the ethylene signal, whereby involved in the hormone regulation of grape growth and development. Furthermore, the promoter of *VvGA3ox2* also contained multiple hormone-responsive elements (Figure 4A), indicating that *VvGA3ox2* might be one potential key node in the progress of multi-hormones interactions.

Based on our preliminary trial of *VvMIR3633a* and *VvGA3ox2* in response to GA, we further performed in vitro validation of their promoter in responses to exogenous GA. Based on binary vector pBI121, the 35SCaMV promoter in pBI121 was replaced with the promoters of *VvMIR3633a* and *VvGA3ox2*, respectively, and fused with the *β-glucuronidase* (*GUS*) reporter gene (Figure 4B). The 35SCaMV promoter in the pBI121 vector was used as a positive control, and the pBI101 vector without a 35SCaMV promoter was used as a negative control. The obtained vector was independently transformed into tobacco plants by an Agrobacterium-mediated method. The different concentrations of GA, including 0, 30, and 50 μM, were applied to the transgenic tobacco lines to verify the response of *VvMIR3633a* and *VvGA3ox2* promoters to GA. As shown in Figure 4C, the positive control of the *GUS* gene was stained to the same extent under control and GA-treated conditions. Similarly, GUS staining driven by the *VvMIR3633a* and *VvGA3ox2* promoters was observed in untreated samples (0 μM GA) of transformed tobacco leaves. Moreover, *VvMIR3633a* and *VvGA3ox2* promoters with GUS gene stained more deeply in transgenic tobacco leaves treated with 30 and 50 μM GA, proving that they might respond to exogenous GA to some extent depending on the content of exogenous GA.

### 2.5. VvmiR3633a-Directed Cleavage on VvGA3ox2

Based on the prediction of *VvGA3ox2* targeting by VvmiR3633a, RLM-RACE technology was used to verify the cleavage role of VvmiR3633a on *VvGA3ox2* in grape berries. RLM-RACE results showed that *VvGA3ox2* was cleaved by VvmiR3633a, the cleavage site was located between the 10th and 11th positions at the end of miRNA5’, and the cleavage frequency was 9/17 (Figure 5A). It confirms VvmiR3633a-directed cleavage of *VvGA3ox2* and leads to a significant repressive effect in grape berries.

The interaction between VvmiR3633a and *VvGA3ox2* was further examined in vivo using tobacco transient transformation technology. The DNA fragment contained VvmiR3633a precursor and cloned into pBI121 to construct the *35SCaMV::VvMIR3633a*. The upstream fragments of the target cleavage site belonging to the gene were fused with *GUS* to obtain *35SCaMV::VvGA3ox2-GUS* and *35SCaMV::mVvGA3ox2-GUS* constructs, respectively. *35ScaMV::GUS* (pBI121) construct was used as a positive control (Figure 5B). We observed a high level of GUS activity in infiltrated leaves with *35SCaMV::VvGA3ox2-GUS* and *35SCaMV::mVvGA3ox2-GUS*. While the infiltrated leaves with *35SCaMV::VvMIR3633a* did not show GUS activity. Furthermore, the transient expression of leaves with *35ScaMV::VvGA3ox2-GUS*+*35SCaMV::VvMIR3633a* led to a significant decrease in the *GUS* expression. These results indicated that VvmiR3633a inhibits the expression of GUS in leaf with *35SCaMV::VvGA3ox2-GUS* through the covalent base pair linking (Figure 5C,D). All these results confirmed that *VvGA3ox2* was directly cleaved by VvmiR3633a, and the former is being the real target of the latter.

### 2.6. Repression of Ovule Development through Exogenous GA Enhancing the Negative Regulation of VvmiR633a on VvGA3ox2 during Grape Embryo Abortion

Here, it was observed that VvmiR3633a had the lowest expression at berries at 30 DAF but the highest at ovules at the corresponding stage (Figure 6A,B), indicating that its modes of action could be different in these two organs above. Moreover, *VvGA3ox_2_* showed similar expression moles to VvmiR3633a during berry and ovule development. Unlike usual negative regulatory modes, VvmiR3633a and *VvGA3ox_2_* had some extent, positive correlations in their expression trends during these two development processes above, implying that their regulatory mechanisms during berry and ovule development can be complicated (Figure 6A,B). Furthermore, the GA responsive expression modes of VvmiR3633a and *VvGA3ox2* were analyzed. Interestingly, we found that GA had no effect on their expression during berry development, while GA remarkably promoted the expression of VvmiR3633a and strongly repressed *VvGA3ox2* expression during ovule development (Figure 6A,B), leading to the enhancement of the negative regulation of VvmiR3633a on *VvGA3ox2* at corresponding development, and suggesting that this module differently behaves in response to GA in various organs, and they might be involved in the modulation of ovule development mainly by responding to GA.

To further recognize the regulatory modes of VvmiR3633a on *VvGA3ox2*, we analyzed the correlation between their expression levels during berry and ovule development. The results showed that in control, both exhibited, to some extent, positive correlation during the development of berry and ovule, with correlation coefficients of 0.3817 and 0.9973, respectively (Figure 6C,D). Interestingly, GA changed the regulatory mode of VvmiR3633a on *VvGA3ox2* during berry and ovule development. It was supported by the fact that both correlation coefficients of VvmiR3633a and *VvGA3ox2* at their expression levels were −0.4296 and −0.8278, respectively, during berry and ovule development in GA-induced ovule abortion. Therefore, GA greatly enhanced the negative regulatory roles of this module during berry and ovule development, especially in the ovule compared to control. Furthermore, we found that exogenous GA remarkably promoted the expression of VvmiR3633a and decreased the expression of *VvGA3ox2*, whereby the endogenous GA synthesis is repressed during the exogenous GA-induced ovule abortion process. It indicates that exogenous GA might repress grape ovule development by boosting the negative regulatory roles of the VvmiR3633a-*VvGA3ox2* module, whereby down-regulation of *VvGA3ox2* level inhibits GA synthesis by up-regulation of VvmiR3633a level.

### 2.7. Expression Profiling of GA-Catabolic Enzymes during Grape Ovule Abortion Process

To create a more comprehensive understanding of the GA-mediated network during modulation of the grape ovule abortion process, here we further compared the spatial-temporal expression profiles of major gibberellin catabolic enzymes, including different isomers of GA-oxidases during this process (Figure 7, Appendix A). It was observed that among GA synthesis enzymes, *VvGA3ox2* and *VvGA20ox2* were highly expressed in ovules at 30 d. Interestingly, the long-term effects of exogenous GA showed the remarkable inhibition of their expression levels at the whole ovule development. Meanwhile, the main promotion of in-active GA genes *VvGA2ox3* and *VvGA2ox1* levels and the remaining members were inhibited, indicating that there was a feedback regulation of exogenous GA in active GA accumulation content, similar to a previous report [30]. Therefore, these 4 genes, including *VvGA3ox2*, *VvGA20ox2*, *VvGA2ox3,* and *VvGA2ox1,* might be the main factors affecting active GA contents during the GA-induced grape ovule abortion process.

### 2.8. GA Down-Regulated the Antioxidant Enzymes during Grape Seed Embryo Abortion

Antioxidant enzymes play important roles during the ovule abortion process. However, it is unclear how exogenous GA affects antioxidant enzyme activity and expression profiles in the modulation of ovule abortion. We analyzed the activity of antioxidant enzymes (VvSOD, VvCAT, and VvPOD) during the GA-induced grape seed embryo abortion process and revealed the long-term effects of exogenous GA treatment except for VvPOD activity that was not detected. VvSOD enzyme activity slowly increased and reached the peak at 34 d, and then decreased during seed development. In contrast, the entire level of GA repressed VvSOD activity during this process, especially at 34 d, decreased at the ebb (Figure 8A). Interestingly, in the control, VvCAT activity showed the typical upward tendency during grape seed development, while exogenous GA remarkably inhibited the VvCAT activity, decreased to the lowest level in GA-induced grape ovules at 31 d, and then remained stable during abortion. These results suggest the reduction of VvCAT and VvSOD enzyme activities might be one of the main reasons for the GA-induced grape ovule abortion process.

Based on the above results, we further searched *VvCAT* and *VvSOD* gene sequences in grapes (Appendix A) and detected related gene expression profiles during seed development (Figure 8B). It was found that some members belonging to the *VvSOD* family, including *VvFeSOD*, *VvFeSOD3*, *VvCu/ZnSOD,* and *VvCuSOD* genes, had high expressions during two stages of grape seed development, except for *VvMnSOD* gene, which contributed to VvSOD protein accumulation for the normal seed development. Similarly, the members of the *VvCAT* family, including *VvNADPH-E*, *VvFeRO2,* and *VvFeRO7* genes, indicated high expression at the least two stages, thereby leading to VvCAT protein accumulation. Importantly, exogenous GA obviously down-regulated *VvFeSOD*, *VvFeSOD3,* and *VvMnSOD* levels of the *VvSOD* family, and *VvNADPH-E* and *VvFeRO2* expressions of the *VvCAT* family, which might be the main factors that lead to the reduction of VvSOD and VvCAT activities as shown in Figure 8A. Therefore, exogenous GA might inhibit the activities of VvCAT and VvSOD mainly by down-regulating the expressions of *VvFeSOD*, *VvFeSOD3*, *VvMnSOD*, *VvNADPH-E*, and *VvROFe7*, which might be one of the important reasons for GA-inducing grape ovule abortion.

### 2.9. GA-Mediated VvmiR3633a-VvGA3ox2 Module and Related Genes Regulatory Network during Grape Ovule Abortion Process

Based on all expression analysis results, we preliminarily deduced a regulatory network pattern in the process of GA-induced grape seed-embryo abortion (Figure 9), where exogenous GA mainly represses the expressions of key enzymes involved in the GA synthesis like *VvGA3ox2* and *VvGA2ox2*, and up-regulates other isoforms of genes that contribute to an inactive form of GA like *VvGA2ox1* and *VvGA2ox3*, whereby leading to reduction of active GA content. On the other hand, exogenous GA inhibits the activity of antioxidant enzyme genes *VvMnSOD*, *VvFeSOD*, *VvFeSOD3*, *VvNADPH-E,* and *VvFeRO2* by decreasing corresponding genes expression levels. Collectively, these two aspects might be the important reasons for causing grape seed-embryo abortion.

## 3. Discussion

The effects of exogenous GA application on the induction of flowering development, seedless berry, and promoting berry enlargement and ripening in grapevines have been widely reported [29,31]. However, the molecular mechanism by which GA-signaling mediates seedless berries formation in grapevines is still not fully understood. In recent years, miRNAs have been identified as key regulatory factors of gene expression through indirect, direct transcriptional, and post-transcriptional gene silencing to modulate the activity of the gene network underlying various developmental and stress-responsive programs [32,33,34]. In Arabidopsis, exogenous GA could promote the miR159 level and delay flowering time and another development, as well as reduce flower fertilization rate [35]. Our previous study revealed that VvmiR159c and VvmiR160s are the key regulators of flower development during GA-induced grape parthenocarpy [29,36]. miR160 targeting of ARF10/16/17 is indispensable for various aspects of AUX-mediated floral organ abscission, as well as ovary patterning in tomato plants [37]. Moreover, other microRNA families, such as miR166, miR156, and miR172, have an irreplaceable role in plant flower and fruit development of Arabidopsis, tomato, poplar, and other plants [38,39,40]. These researches demonstrated that miRNAs are important regulators in plant flower and fruit development, including ovule development.

Similarly, our previous high-throughput sequencing of GA-induced grape parthenogenesis identified the exact sequence of a GA responsive VvmiR3633a in ‘Fujiminori’ grape berries [27], which is identical to ‘Pinot Noir’ in miRBase 21.0, and predicted its potential target gene *VvGA3ox2*, a key enzyme in GA synthesis pathway, together with the typical effect of GA application in the induction of grape seed-embryo abortion. Thus, we confer the VvmiR3633a-*VvGA3ox2* module to be involved in the modulation of this process and our current work validated this hypothesis. Here, the application of exogenous GA on the ‘Fujiminori’ grape berries could effectively induce seed-embryo abortion with a seedless rate up to 98.96% 2 days after flowering. Furthermore, we further employed the 5′-RLM-RACE technique to validate the cleavage role of VvmiR3633a on *VvGA3ox2* in grapes, with having one cleavage site at the complementary target region of the 10th position from the 5′-end of VvmiR3633a. This cleavage site is one of the main sites where miRNAs cleave their target genes, supported by the previous studies that the cleavage sites of miRNAs on their targets mainly occurred at the 9th, 10th, and 11th from the 5′-end of the miRNA sequence [41,42]. In addition, the interaction between VvmiR3633a and *VvGA3ox2* was further verified in vivo by the Agrobacterium-mediated tobacco co-transformation technique. All these results confirmed the cleavage role of VvmiR3633a on *VvGA3ox2*, and the latter is one true target of the former.

Gibberellin 3β-hydroxylase in plants is a class of Fe(II)-2OG dioxygenases that catalyze inactive GA to functionally active GA [43]. The catalytic activity of GA3ox mainly depends on two conserved domains of 2OG-FeII Oxy and DIO N, which are important sites for gibberellin 3β-hydroxylase catalysis [13]. Our analysis determined that 21 species, including grapes, contained 2OG-FeII Oxy and DIO N domains, indicating a high degree of conservation of GA3ox across various species. Although the phylogenetic tree divides GA3ox proteins into three groups, the genetic domain sequences (except A and B) and the protein motif are highly similar, also demonstrating the high conservation of GA3ox evolution across different species.

In this study, the application of exogenous GA 2 days after flowering caused grape seed-embryo abortion, accompanied by a drastic increase in the expression level of VvGA3ox during ovule development, which led to a sharp reduction in VvGA3ox levels and promoted seed-embryo abortion. Interestingly, during berry development, the VvmiR3633a-*VvGA3ox2* module exhibited a different response pattern to GA than ovule development, where this module was hardly affected in their expressions by exogenous GA (Figure 6). These results indicated that the VvmiR3633a-*VvGA3ox2* module possesses tissue-specific expression mode in grapes, implying the functional diversity of this module in various tissues or organs. More importantly, exogenous GA significantly changed the regulatory modes of VvmiR3633a on *VvGA3ox2* during berry and ovule development according to the positive correlation in control samples, which were 0.3817 and 0.9973 for berry and ovule, respectively; and negative correlation in treated samples with GA treatment that were −0.4296 and −0.8278 for berry and ovule, respectively. Therefore, the shift in ovules is stronger than that in berries. In addition, we also discovered that in normal berries and ovules without GA treatment, the VvmiR3633a down-regulated in berries during the stone-hardening stage (at 30 d), but on the contrary, the expression peaked in ovules without GA treatment. We also observed a similar pattern of expression regarding its target gene *VvGA3ox2* during corresponding periods, suggesting VvmiR3633a-*VvGA3ox2* module is a stage-specific trait during grape berry and ovule development. Especially, the expression peak of *VvGA3ox2* in normal ovules at 30 d promotes the accumulation of active GA, which might be one of the potential pathways of seed-producing active GA. All these results further suggest the functional diversity of the VvmiR3633a-*VvGA3ox2* module, and they might play important roles in GA signal regulation of grape seedless berry development.

## 4. Materials and methods

### 4.1. Plant Material and GA Treatment

We used a variety of grapes called “Fujiminori,” with bright color, plump grains, and easy storage and transportation for early preliminary experiments and horticultural practice. Samples were collected from a 5-year-old tree and the experimental time was from May to August 2018 at the grape base of Jiangsu Academy of Agricultural Sciences. The base is cultivated in steel pipe multi-span greenhouses. The spacing of the planting row was 3 m, and the dwelling was 1.5 m.

Based on the preliminary experiment and horticultural practice, young berries were impregnated with 50mg /L GA (supplemented with 0.05% Tween-20) for 30 s at 2 days after flowering. Control clusters were treated using water with the same process. Samples were collected at the following time points; 5, 15, 25, 28, 31, 34, 37, 40, 43, and 47 days after flowering, frozen rapidly in liquid nitrogen, and stored at −80 °C for the next analysis.

### 4.2. RNA Extraction, Low Molecular RNA Isolation, and cDNA Synthesis

Total RNA from each sample was extracted by the Cetyl trimethyl ammonium bromide (CTAB) method as described by Zhang [36]. The separation of the low molecular weight (LMW) RNA and high molecular weight (HMW) was performed using 4M LiCl. Then as described by Wang [28], LMW RNA was polyadenylated at 37 °C for 60 min in a 50-μL reaction mixture with 1.5 μg of total RNA, 1 mM ATP, 2.5 mM MnCl2, and 4 U poly(A) polymerase (Ambion, Austin, TX, USA). Poly(A)-tailed small RNA was recovered by phenol/chloroform extraction and ethanol precipitation. 5′ adapters (5′-CGACUGGAGCACGAGGACACUGACAUGGACUGAAGGAGUAGAAA-3′) were ligated to the poly(A)-tailed RNA using T4 RNA ligase (Invitrogen, Carlsbad, CA, USA). These LMW RNAs with Poly (A) and 5′-adapter were reverse transcribed into cDNAs. Finally, the reverse transcriptase was inactivated by incubation for 15 min at 70 °C, and cDNA for mRNA and poly(A)-tailed small RNA were stored at −80 °C.

### 4.3. Cloning and Identification of VvmiR3633a Precursor and Mature Sequence

According to the method reported by Zhang [28], the mixtures of diverse stage LMW RNA samples were loaded into Poly(A) tails using a Poly(A) Tailing kit from TAKARA. The Poly(A)-tailed LMW RNAs were further ligated to a 5′ adapter (5′-CGACUGGAGCACGAGGACACUGACAUGGACUGAAGGAGUAGAAA-3′) using T4 RNA ligase (Invitrogen, Carlsbad, CA). These LMW RNAs with Poly (A) and 5′-adapter were reverse transcribed into cDNAs for clones of VvmiR3633a sequences by miR-RACE, which we previously developed [41].

### 4.4. Prediction, Cloning, and Subcellular Localization of VvmiR3633a Target Gene

Based on the precise sequences of VvmiR3633a, we employed Multiple Align (BioXM software) to align the sequence. The miRNA unique sequences were further used as the benchmark to perform a blast search in grape transcript database V2 (http://genomes.cribi.unipd.it/grape/) (accessed on 5 September 2018) to predict the target genes of VvmiR3633a. According to the method reported by Song [44], the mixtures of total RNAs at different stages were reverse-transcribed into cDNAs for cloning the target gene. The full-length *VvGA3ox2* coding region was amplified using specific primers. The fragment was restricted and ligated to the binary vector pCAMBIA1302 with the modified green fluorescence protein (GFP) gene driven by the CaMV35S promoter. The construct was mobilized into the *Agrobacterium tumefaciens* strain EHA105α by the freeze-thaw method. Leaf infiltration of tobacco with bacterial suspension was performed using a needleless syringe. The infiltrated plants were kept in the culture room for 2 days. Leaf portions were observed under LEICA TCS SP5 confocal microscope (Germany) using GFP channels to localize the expressed 35S-*VvGA3ox2*:GFP fusion and 35S-GFP proteins.

### 4.5. Amino Acid Sequence Analysis of VvGA3ox2 and Its Homologous Protein

Homologs proteins of *VvGA3ox2* were identified by blast searching using nucleotide and amino acid sequences of *VvGA3ox2* in the NCBI databases. Then, the Clustal X2 and DNAMAN software were used to analyze the amino acid sequences of *VvGA3ox2* and its homologous protein. The MEME 5.05 online program (http://meme-suite.org/tools/meme) (accessed on 13 September 2018) was used to analyze the motif composition as described by Zhang [28].

### 4.6. Phylogenetic Analysis

The protein sequences of *VvGA3ox2* and its homologous protein were imported into MEGA5 after multiple alignments using ClustalX2 in this software. The unrooted phylogenetic trees were constructed using the neighbor-joining (NJ) method and bootstrap tests were carried out for 1000 iterations.

### 4.7. Analysis of Motif Elements of the Promoters from VvMIR3633a and VvGA3ox2 Promoter

The promoters of *VvMIR3633a* and *VvGA3ox2* (approximately 1500 bp upstream of genes) were obtained from the grape gyroscope database (http://www.genoscope.cns.fr/externe/GenomeBrowser/Vitis/) (accessed on 15 September 2018) and PlantCARE database was used to predict motif elements in the promoter regions of genes.

### 4.8. Expression Analysis of VvmiR3633a and Its Target Genes during Berry and Ovule Development by qRT-PCR

The template for qRT-PCR was the cDNA of poly(A)-tailed small RNA, as mentioned above. To amplify VvmiR3633a from reverse-transcribed cDNAs, we used the precise sequences of VvmiR3633a as the forward primer (AAACTGGAAAGCTGGCATGG) and R16328 (ATTCTAGAGGCCGAGGCGGCCGACATG) as the reverse primer as previously described by Wang [28]. The qRT-PCR templates for *VvGA3ox2* belonged to the HMW-enriched library, which was used for qRT-PCR amplification. *VvActin* was used as a reference gene to normalize qRT-PCR data. The relative expression level was calculated with the formula 2^−^^ΔΔCT^ normalized expression ratio [29]. Each PCR assay was carried out with three biological replicates, and each replicate corresponded to three repeats of separate experiments. All primers were listed in Additional Appendix A.

### 4.9. Mapping of mRNA Cleavage Sites Using RLM-RACE and PPM-RACE

To map miRNA-mediated cleavage products, RNA ligase-mediated rapid amplification of 5′ cDNA ends (RLM-RACE) and poly(A) polymerase-mediated 3′ rapid amplification of cDNA ends (PPM-RACE) were employed in our work [28]. All data were normalized using *VvActin* as a reference gene. Each PCR assay was carried out in three biological replicates, and each replicate corresponded to three technical repeats of separate experiments.

### 4.10. Construction of the Expression Vector

The promoter sequences of *VvMIR363a* and *VvGA3ox2* were isolated. The used primers were described in Appendix A. The DNA fragment containing the VvmiR3633a precursor was cloned into pBI121 to make a *35SCaMV::VvMIR3633a* construct. The target gene *VvGA3ox2* and the upstream fragments of the *VvGA3ox2* target gene cleavage site were fused with *GUS* to obtain *35SCaMV::VvGA3ox2-GUS*, and *35SCaMV::mVvGA3ox2-GUS* constructs, respectively. A *35ScaMV::GUS* (pBI121) construct was used as a positive control.

### 4.11. Agrobacterium-Mediated Tobacco Transient Transformation and GUS Staining

Agrobacterium-mediated tobacco transient transformation was performed as described by Zhang [36]. Leaves of 6-week-old tobacco plants were infiltrated with the bacterial suspension by injection. Tiny holes were made in the tobacco leaves using syringe needles and the bacterial suspension was injected using injection needles. The infiltrated seedlings were then moved back to the environmental chamber and kept in the dark for 3 d. The uniformly sized leaves were used in the infiltration experiment, and the experiment was repeated three times.

### 4.12. GUS Assay

Histochemical GUS staining of infiltrated leaves was performed as described by Zhang [36]. Leaves were immersed in ethanol to remove chlorophyll. Quantification of GUS activity was determined using the fluorometric 4-methylumbelliferyl-b-D-glucuronide (MUG) method. One unit of GUS activity was defined as 1 nM of 4-methylumbelliferon (4-MU) generated per minute per milligram of soluble protein. In each independent experiment, three leaves were infiltrated with each construct and combined to detect GUS activity.

### 4.13. Determination of Antioxidant Enzyme Activity

Peroxidase (POD), catalase (CAT), and superoxide dismutase (SOD) activities of embryo samples were determined by kit. The kit was purchased from Beijing Solaibao Technology Co., Ltd. (Beijing, China). The samples were treated and all indicators were determined in accordance with the instructions of the kit, and each sample was repeated 3 times.

### 4.14. Data Analysis

Excel 2019 was used for statistical analysis of data; Origin 2021B and Adobe Photoshop 2021 were used for chart making.

## 5. Conclusions

Collectively, we examined the changes in berry and ovule during the exogenous GA-induced seed-embryo abortion process in the ‘Fujiminori’ grape. We isolated VvmiR3633a and its target *VvGA3ox2* from grape berries and validated the cleavage role of the VvmiR3633a-*VvGA3ox2* module in berries. This module exhibited tissue- and stage-specific traits during berry development in the grape stone-hardening stage. Furthermore, the exogenous GA significantly changed the regulatory model of this module from positive to negative regulation, especially in ovules. GA represses ovule development through a multi-level cascade signal network of the VvmiR3633a-*VvGA3ox2* module and their mediated related genes during the grape seed-embryo abortion process. Our results demonstrated VvmiR3633a-*VvGA3ox2* module plays an important role in GA signal to regulate grape seed-embryo abortion, which has significant implications for the development of the high-quality seedless grape berry.

## Figures and Tables

**Figure 1 ijms-23-08767-f001:**
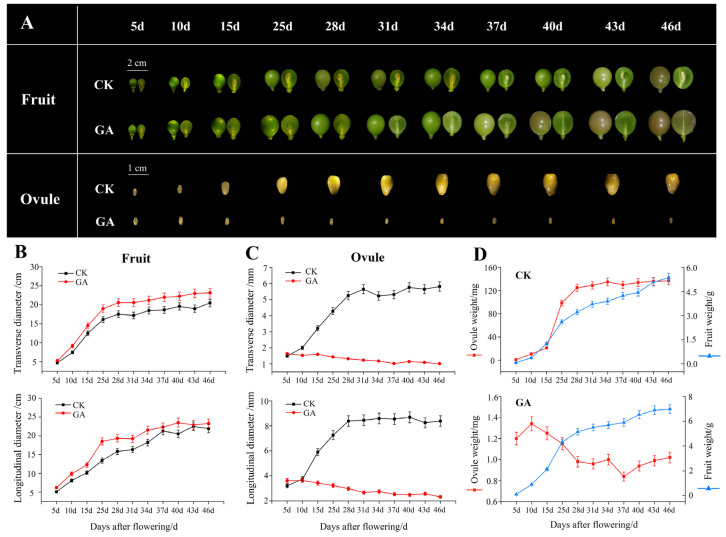
Morphological changes of grape berries and embryos after GA treatment. The appearance of grape and ovule changed after exogenous GA treatment. The appearance of the berry and ovule (**A**), the changes in the transverse and longitudinal diameter of berries (**B**), changes in transverse and longitudinal diameters of ovules (**C**), and changes in berry and ovule weight per grain (**D**) in response to GA exogenous. d in horizontal axes represents days after grape flowering.

**Figure 2 ijms-23-08767-f002:**
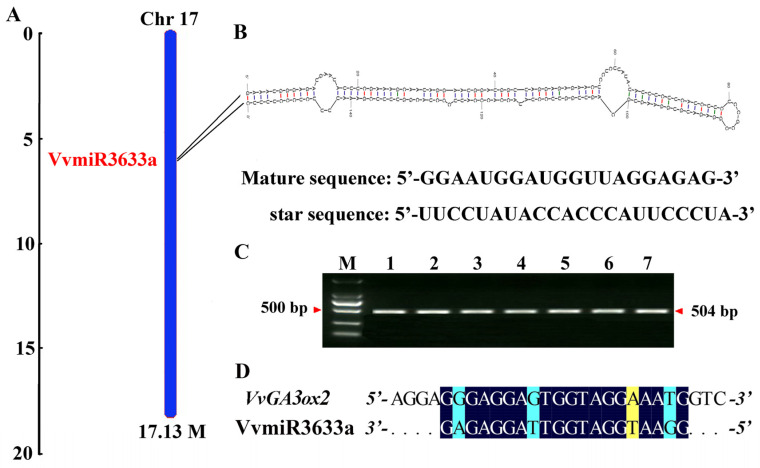
Identification of microRNA3633a and its target genes. Chromosomal location of VvmiR3633a (**A**); The stem-loop structure of pre-VvmiR3633a and its mature sequence and star sequences (**B**); Cloning of *VvMIR3633a* gene (**C**); The interaction pattern and mismatch rate of VvmiR3633a and its *VvGA3ox2* target genes (**D**).

**Figure 3 ijms-23-08767-f003:**
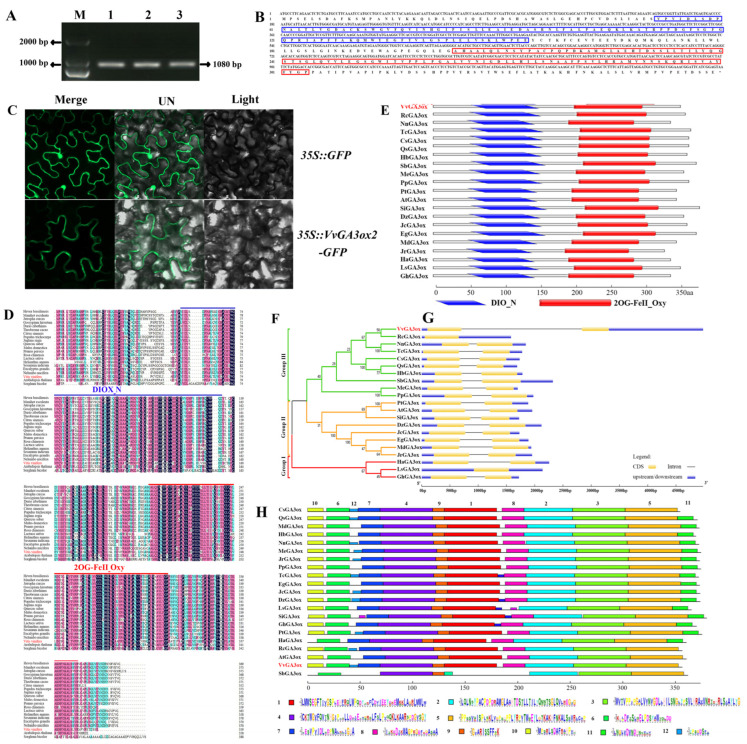
Sequence analysis and subcellular localization of target genes. Cloning of *VvGA3ox2* (**A**); *VvGA3ox2* amino acid sequence deduction (**B**); Subcellular localization (**C**); Alignment of GA3ox2 amino acid sequence homology and conserved domains in different species (**D**,**E**); GA3ox2 evolutionary analysis (**F**); GA3ox2 gene structure analysis (**G**); Analysis of Conserved Motifs of GA3ox Protein Sequences in Different Species (**H**).

**Figure 4 ijms-23-08767-f004:**
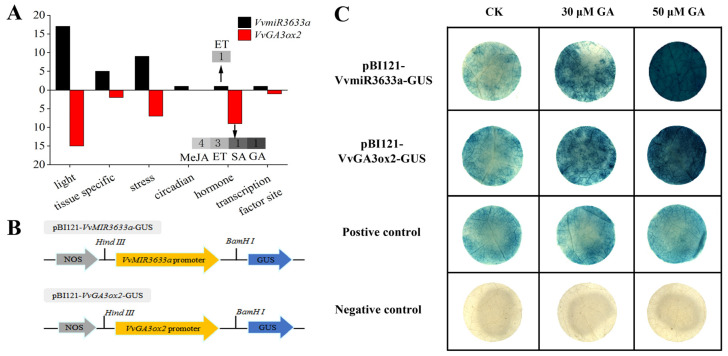
Classification of cis-acting elements in promoter regions (**A**) and promoter activity of *VvMIR3633a* and *VvGA3ox2* in response to GA (**B**,**C**).

**Figure 5 ijms-23-08767-f005:**
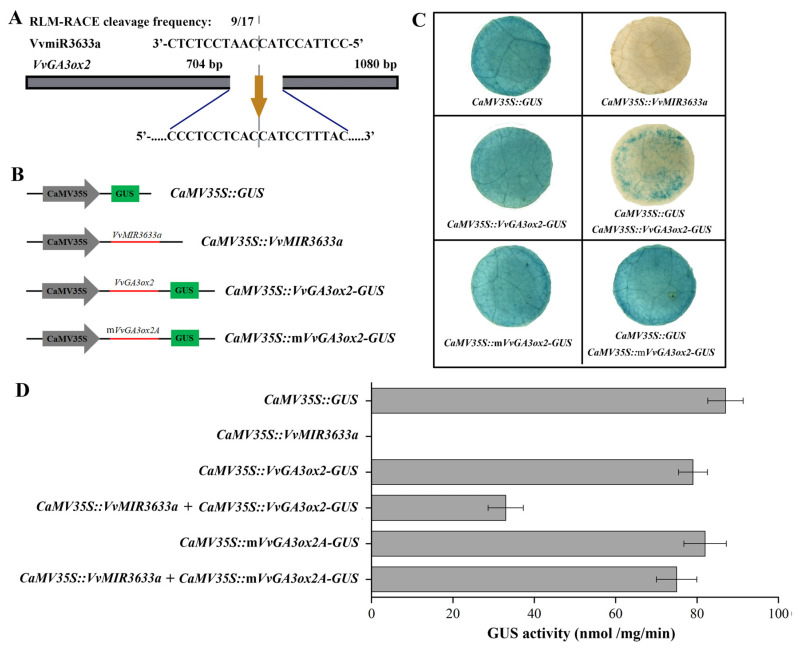
Validation of target genes cleavage by VvmiR3633a. The cleavage of *VvGA3ox2* by VvmiR3633a was verified by the RLM-RACE technique (**A**) and tobacco transient expression technique (**B**,**C**); GUS activity (**D**).

**Figure 6 ijms-23-08767-f006:**
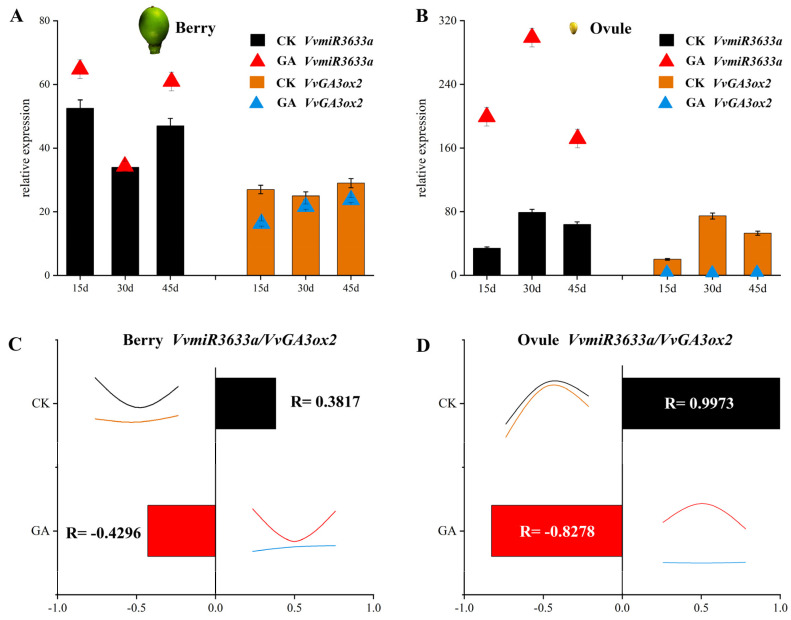
Expression analysis and correlation of miR3633a-GA3ox2 module in response to GA in berries (**A**,**C**) and ovules (**B**,**D**).

**Figure 7 ijms-23-08767-f007:**
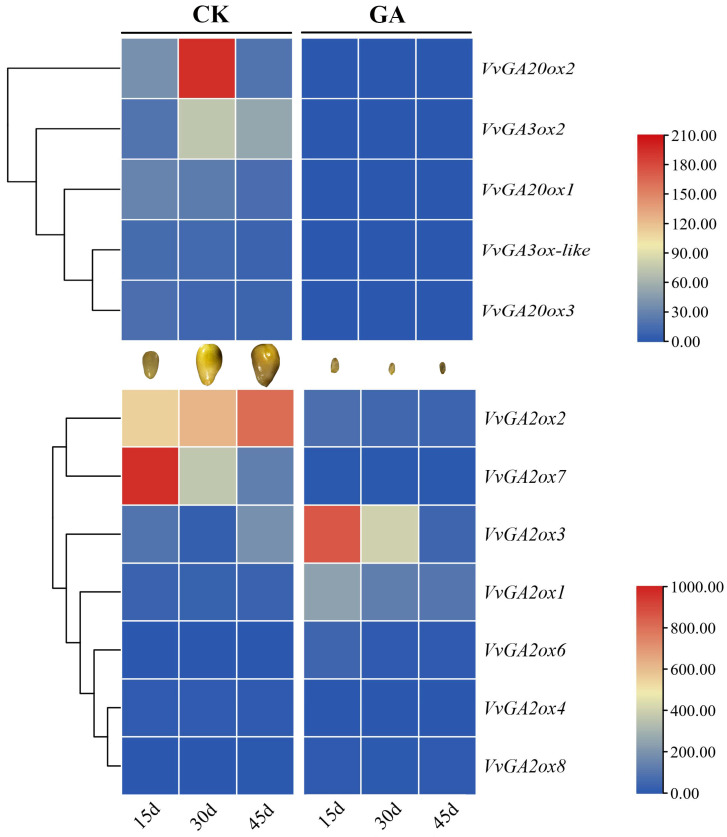
Expression modes of gibberellin catabolic enzymes according to time course experiment during grape and seed development.

**Figure 8 ijms-23-08767-f008:**
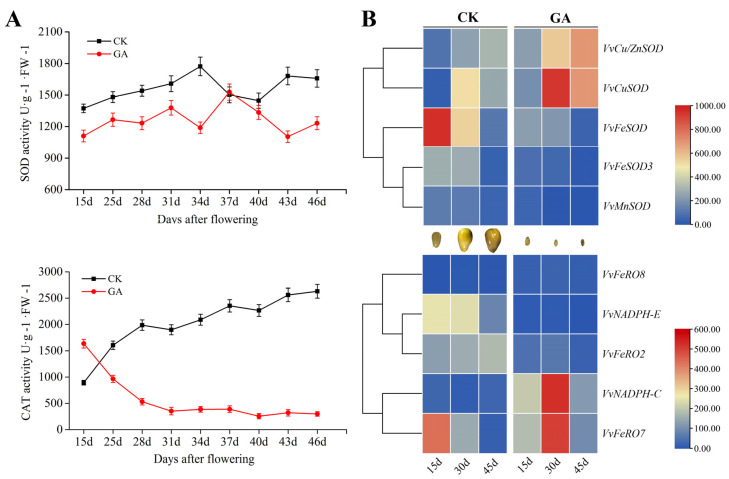
The activity of SOD and CAT enzymes (**A**) and the expression profile of members belonged to SOD and CAT family (**B**) during seed development in response to GA.

**Figure 9 ijms-23-08767-f009:**
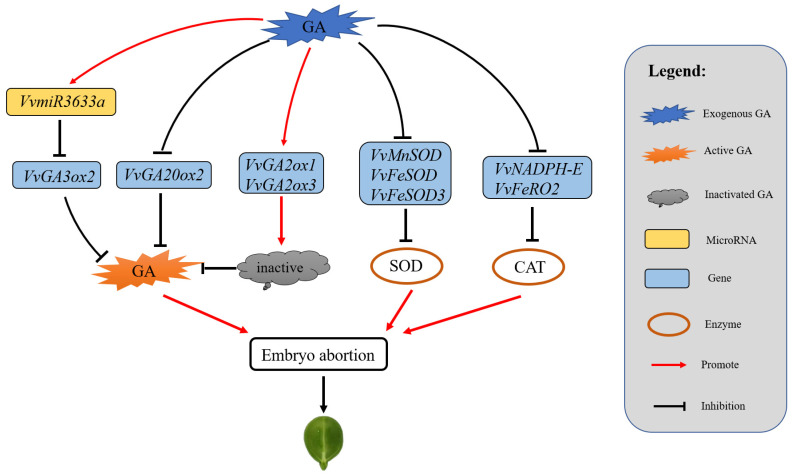
Regulation network of grape embryo abortion induced by exogenous GA.

## Data Availability

All data generated or analyzed during this study are included in this published article.

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
