# Peer review of "miR3633a-GA3ox2 Module Conducts Grape Seed-Embryo Abortion in Response to Gibberellin"

_ijms, 2022, doi:10.3390/ijms23158767_

Round 1

Reviewer 1 Report

For Introduction

Lines 81-82: Related to: ... <and the related mechanism is also unclear yet even in model plants like Arabidopsis, poplar, and rice>.

 I think you should mention the popular name for Arabidopsis, or mention the scientific names at all. I would still suggest the scientific names. I think you are referring, not only to species of the genus Arabidopsis, but to a specific species, that is Arabidopsis thaliana, which is known as a model used in laboratory research.

For Results

Line 93: What is "Fujiminori"? A variety of grapes? It is not clear. Please detail or specify for the reader to understand without looking for information in other sources.

Line 104: What does 5d, 10 d.....46d from Figure 1 A B C D represent? Are there days? Then you should at least make a reference to them in the text. You talk about weeks, but in the figure there are days. I think you have to relate to what is in the figure.

Lines 93-103: I find the presentation of Figure 1 too brief. Detail the paragraph considering the data from the graphs in Figure 1.

Line 147: The correct name is <Rosa chinensis>

Line 152: Idem: The correct name is <Rosa chinensis>

For Materials and Methods

At 4.1. Plant material and GA treatment

Lines 407-414: Please place the experiment in time. What is the period in which you did the study? In what year(s)? For example: June-August, 2021 or something like that? There is no information on the climatic conditions in the culture area, so it is necessary to ensure the minimum amount of information related to the time of a year.

For Conclusion

Line 521: I think it's a typo related to <Fujinminor>. Probably <Fujiminor> is correct because in the mentions above you say so.

Author Response

Dear Reviewer,

We have uploaded the response as a word document.

Best regards Bunhe Bai

Reviewer 2 Report

In the MS, the AUTHORS have given an exhaustive account miR3633a-GA3ox2 module conducts grape seed-embryo abortion in response to gibberellin. The authors have put in a good amount of effort, and the work is appreciable. This paper deals with an important and interesting topic. The authors present their data in a novel and comprehensible manner, making it an interesting subject for researchers. Overall, I consider that the paper adopts new research methods and makes an excellent contribution to this research field; I recommend article to accept for publication after some minor revision. For the better readability of the manuscript, I would like to recommend some changes and improvements.

1.      First time, mention a complete name and later use an abbreviation. Kindly thoroughly check the whole MS. I have seen in the MS that the author used e abbreviation many times.

2.      There are several mistakes in the whole manuscript; please add or remove extra characters.

Author Response

Dear Reviewer,

We have uploaded the response as a word document.

Best regards Yunhe Bai
